# Use of a Satellite-Based Aridity Index to Monitor Decreased Soil Water Content and Grass Growth in Grasslands of North-East Asia

**Reiji Kimura [1,*] and Masao Moriyama [2]**

[1]   Arid Land Research Center, Tottori University, Tottori 680-0001, Japan
[2]   Graduate School of Engineering, Nagasaki University, Nagasaki 852-8521, Japan; matsu@nagasaki-u.ac.jp
*   Correspondence: rkimura@tottori-u.ac.jp; Tel.: +81-857-21-7031

**Abstract:** Numerous simulation studies of the effect of global warming on arid regions have indicated that increases in temperature and decreases in precipitation will trigger water shortages, drought, and further aridification. In north-east Asia, especially China and Mongolia, the area of degraded land has increased since 2000. Land use in arid regions is mainly natural grasslands for grazing. Growth in this land use is limited by the precipitation amount and intensity. To develop sustainable management of grasslands, it is essential to examine the relationship between water consumption and the growth patterns of the grasses. This study examined the applicability of a satellite-based aridity index (SbAI) as a way to measure the water consumption and growth of grasslands in China and Mongolia. The effective cumulative reciprocal SbAI was strongly correlated with the cumulative decreased soil water content in the root zone and changes in the normalized difference vegetation index in Shenmu, China. Application of the effective cumulative reciprocal SbAI to grasslands in Mongolia and in north-east Asia revealed a high correlation between the effective cumulative reciprocal SbAI and changes in the normalized difference vegetation index (NDVI). The effective cumulative reciprocal SbAI might be suitable for the detection of water consumption and growth in grasslands from satellite data alone.

**Keywords:** desertification; drylands; remote sensing; thermal inertia

## 1. Introduction

Drylands are highly vulnerable to climate change. In particular, increasing global warming can increase temperatures and decrease precipitation in drylands at high latitudes [1–3], and their effects on ecosystems can increase [4,5]. Drought has been increasing in Mongolia since 2000 [6,7]. The main land use in the arid regions of north-east Asia, especially in China and Mongolia, is natural grasslands, and the main source of livelihood is stock grazing on grasslands. However, the growth of natural grasslands depends on unstable rainfall [8–10]. The coverage of natural grasslands has a profound effect on the occurrence of dust outbreaks [11–13], which bring damage to agriculture, stock grazing, and human and animal health, not only locally but also in neighboring regions such as Japan and Korea [14–16]. An early warning and monitoring system based on numerical models, remote sensing, and weather forecasts is urgently needed to guard human well-being and to control natural hazards in those regions. Such models must take into account the growth of vegetation (in this case, grasslands).

Numerical models of vegetation growth are different in climatic features [17]. In humid regions like Japan where precipitation is abundant (1718 mm average from 1971 to 2000: [18]), the quantity of growth depends on the temperature. In arid regions with an adequate sunshine duration and temperature, however, it depends strongly on precipitation and thus soil water content (SWC). Changes

in the normalized difference vegetation index ($\Delta_{NDVI}$) were closely related to accumulated rainfall and decreased SWC in natural grasslands in China [19]. Thus, observed values of rainfall and SWC as model inputs will affect the accuracy of numerical simulations of plant growth.

However, meteorological data are limited in arid regions, especially rainfall, leading to uncertainties in the simulated values of SWC. With a high resolution and frequency, satellite data offer advantages in monitoring vegetation growth in arid regions [20]. For example, the Moderate Resolution Imaging Spectroradiometer (MODIS) and Copernicus Missions (specifically Sentinel 1 or 2) has provided data since 2000 and 2014, respectively. Because lengthy cloudless periods are common in arid regions, much of the MODIS or Sentinel data are usable for global analyses [21]. However, implementation of remote sensing and imagery techniques to monitor and detect the effects of extreme weather events and climate change on vegetation growth is still underdeveloped [22].

In Japan, where precipitation is abundant, the effective cumulative temperature (ECT) is commonly used to simulate vegetation growth [23–25]. ECT is the effective cumulative daily mean temperature above a threshold (e.g., 10 °C in [25]) and is related to plant growth [24]. ECT has been developed into developmental indices, calculated by summing the developmental rate with respect to time [26–28].

Because vegetation growth in arid regions depends on precipitation or SWC, we got the idea from ECT and developmental index methods to replace the temperature index with a water index obtained from satellite data to monitor growth. Products using passive or active microwave sensors, such as the Advanced Microwave Scanning Radiometer–EOS (AMSR-E) and AMSR-2, are used to detect SWC [20,29–32]. However, since their resolution is low (e.g., 50 km in AMSR-2), the accuracy of the soil moisture estimation is low [31,32]. However, the satellite products usable for SWC estimation, from thermal inertia calculated from land surface temperature (LST), have a high resolution and high observation frequency, and give good agreement with observed SWC [33–35], thus offering advantages for monitoring drought, desertification, and dust occurrence [6,36,37].

A satellite-based aridity index (SbAI) was proposed [38], which is an indicator of the thermal inertia calculated from day–night LST differences, to monitor the extent of the aridity or wetness. The SbAI has been validated and used to identify degraded land and drought in north-east Asia and globally [6,36,39]. Here, we used the ECT method, replacing temperature with the SbAI calculated from satellite data, to monitor vegetation growth and consumed SWC in north-east Asia, focusing on the natural grasslands of China and Mongolia.

## 2. Methods

### 2.1. Study Areas and Periods

#### 2.1.1. Shenmu, China

The observation site was an experimental station in the Liudaogou River Basin (7 km$^2$) in Shenmu District, Shaanxi Province, China (38°47′ N, 110°21′ E, 1224 m above sea level; Figure 1) [40]. The average annual temperature from 1957 to 1989 was 8.4 °C, with an average annual rainfall of 437 mm [41]. The region is semi-arid [42].

Data of the SWC in Shenmu, China, were recorded from June 2004 to March 2010, but we used the SWC data from 2005 to 2007, a timespan when the data quality (data within the operating life (=5 years) of the SWC instruments) and a continuous data acquisition (without missing data due to some troubles) could be assured. This period fell during the Chinese Government's Grain for Green Program, intended to reduce the farmland area and promote greening, so the effects of human-managed grazing were low. The station was covered by *Stipa bungeana* Trin., with the root zone within 34 cm of the surface.

The soil texture is a sandy loam (48% sand, 39% silt, and 13% clay). The volumetric SWC was 0.43 m$^3$ m$^{-3}$, field capacity was 0.19 m$^3$ m$^{-3}$, and permanent wilting point was 0.05 m$^3$ m$^{-3}$. The SWC at ten depths (Delta-T; ML2X; 6, 10, 18, 26, 34, 42, 50, 58, 66 and 100 cm) and rainfall (Davis Inst. Corp; Rain Collector 2) were sampled every minute with a datalogger (Campbell Scientific; CR10X), except for the SWC at 12:00 and 24:00 Beijing standard time (BST). A soil moisture sensor was installed vertically

from the soil surface at a depth of 6 cm; therefore, the averaged SWC up to 6 cm was measured. Except for this installation at 6 cm, the sensors were set horizontally. The averaged SWC within the root zone (0–34 cm) was used for analysis. The total rainfall was 308 mm in 2005, 385 mm in 2006, and 481 mm in 2007.

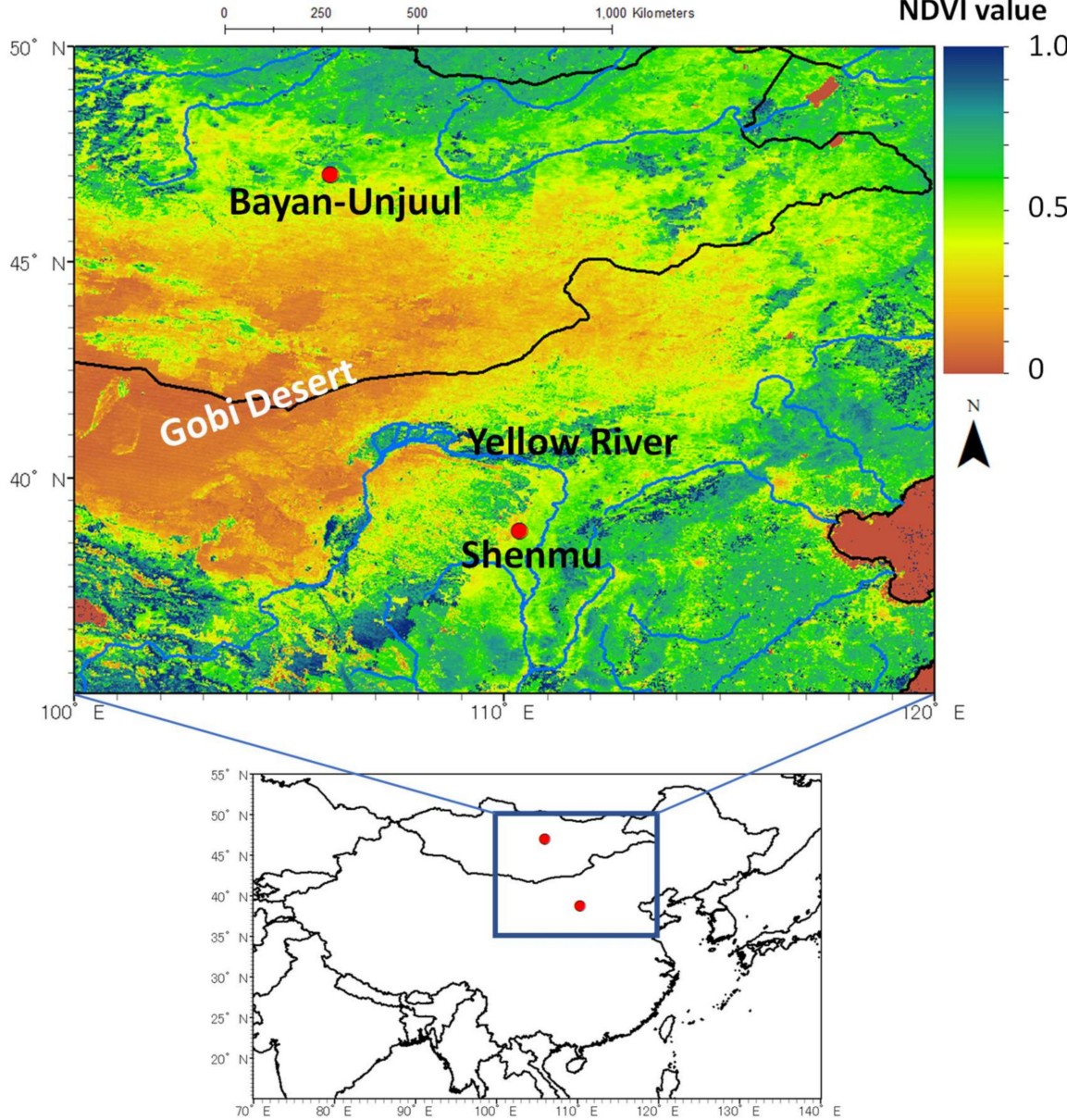

**Figure 1.** Locations of Shenmu, China, and Bayan-Unjuul, Mongolia, with the distribution of the NDVI in August 2019.

### 2.1.2. Bayan-Unjuul, Mongolia

We selected a validation site at Bayan-Unjuul, northern Mongolia (47°02′38.7″ N, 105°56′56.1″ E, 1200 m above sea level; Figure 1), where a 300-m × 300-m fence was erected in June 2004 to exclude livestock [43]. The selected period to validate the proposed method was 2017 to 2019, the most recent three years; 2017 featured a prolonged period of severe dry weather, when an estimated 80% of the country was affected by drought [44], and 2018 featured heavy rainfall [45].

The average annual temperature from 1995 to 2005 was 0.1 °C and the average annual rainfall was 163 mm [43]. Most rainfall is concentrated from May to August (124 mm). The soils are classified as

Kastanozems. The soil texture is sandy loam. The field capacity was 0.20 m$^3$ m$^{-3}$ and the permanent wilting point was 0.05 m$^3$ m$^{-3}$, close to those at Shenmu. The station was covered by short grasses, including *Stipa krylovii* and *Agropyron cristatum*, and the shrub species *Caragana microphylla*. The root zone was limited to 25 cm by a stiff CaCO$_3$ layer (caliche) at that depth [12].

### 2.2. Data and Calculation of the NDVI and SbAI

The NDVI is defined as follows [46]:

$$\text{NDVI} = \frac{\rho_{\text{NIR}} - \rho_{\text{RED}}}{\rho_{\text{NIR}} + \rho_{\text{RED}}}, \tag{1}$$

where $\rho_{\text{NIR}}$ is surface reflectance from MODIS band 2 (841 to 876 nm) and $\rho_{\text{RED}}$ is that from MODIS band 1 (620 to 670 nm).

The 16-day NDVI was calculated with MOD09A1 8-day global surface-reflectance data in the near-infrared and visible (red) wavelength channels at a 500-m resolution. The 8-day NDVI with the highest score (i.e., little influence of cloud, sensor angle, and aerosol concentration) was selected as the 16-day NDVI. The spatial resolution was changed to 1 km by averaging 4 cells. Finally, the monthly NDVI was calculated by averaging the 16-day NDVI.

The SbAI is defined as follows [38]:

$$\text{SbAI} = \frac{\Delta T_\text{s}}{R_\text{s}}, \tag{2}$$

where

$$R_\text{s} = (1 - r) S_0 \cos \theta_\text{c}. \tag{3}$$

Here, $\Delta T_\text{s}$ is the difference in LST between day and night, and $R_\text{s}$ is the absorbed solar radiation calculated from broadband albedo $r$, the solar constant $S_0$ (1367 W m$^{-2}$), and the solar zenith angle at the Sun's apex, $\theta_\text{c}$. $\Delta T_\text{s}$ is an indicator of both surface dryness and other factors, especially solar radiation. To compensate for the effect of solar radiation, $\Delta T_\text{s}$ is divided by $R_\text{s}$ in Equation (2). For a dry surface, SbAI is large because $\Delta T_\text{s}$ is large, and $\Delta T_\text{s}$ is large because the low land surface wetness causes the thermal inertia to be low.

The concept and assumptions of the relationships among SWC, vegetation density, and SbAI are shown in Figure 2a. When the grass density is low, the little rainfall from March to June maintains the SWC in only the surface soil, not the deeper soil [40]. Therefore, SbAI indicates mainly the thermal inertia of the surface soil. When heavy rainfall from July to August infiltrates down to the root zone, the grass density becomes high [40]. Water uptake by the grass roots becomes active, resulting in an increased water content of the grass and thus increased thermal inertia, which is monitored by SbAI.

SbAI was calculated by the method of [38] using the MODIS data products MOD09CMG (daily global surface-reflectance) and MOD11C1 (daily global day/night LST). The spatial resolution of both products was 0.05°. For both the NDVI and SbAI, cells covering each site were used for analysis.

To plot the spatial distribution of the precipitation in north-east Asia, we used the CPC Global Unified Precipitation data provided by NOAA/OAR/ESRL PSL (Boulder, CO, USA; https://psl.noaa.gov/). The spatial resolution was 0.5° × 0.5°.

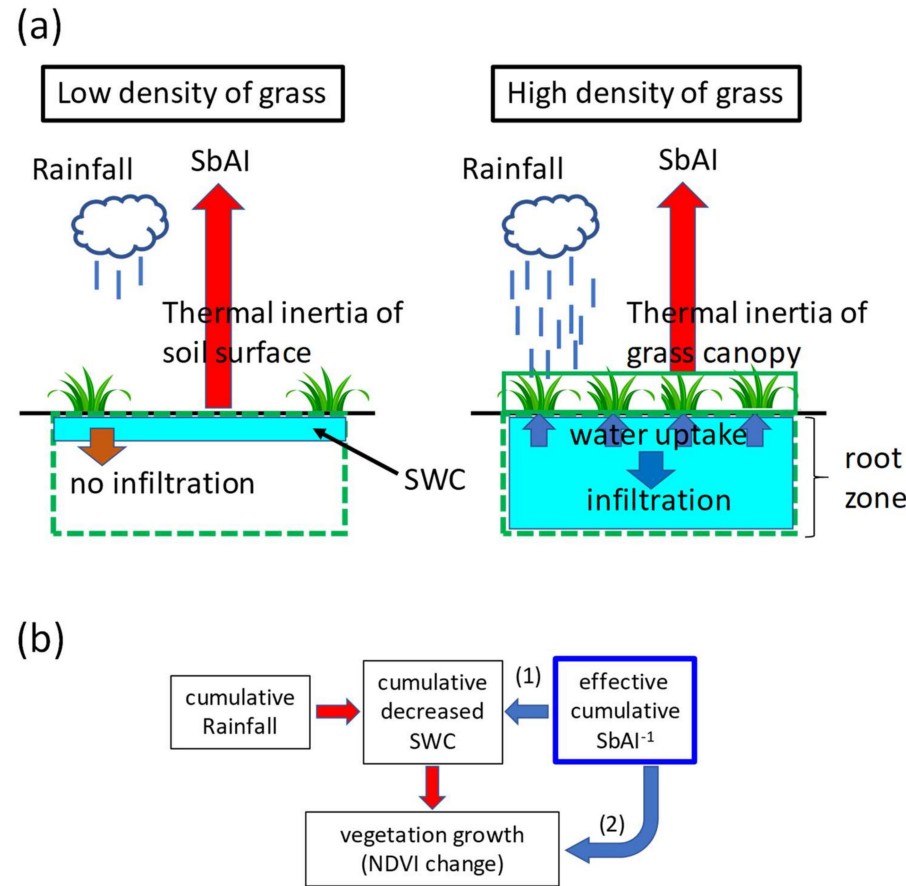

**Figure 2.** Concepts of this study. (**a**) Relationships among the vegetation density, soil water content (SWC), and satellite-based aridity index (SbAI). Red upward arrow, thermal inertia; green dashed box, root zone; green solid box, grass canopy. (**b**) Relationships among the cumulative rainfall, cumulative decreased SWC (CdSWC), effective cumulative 1/SbAI (CRSbAI), and vegetation growth ($\Delta_{\mathrm{NDVI}}$). Red arrows, relationship among cumulative rainfall, CdSWC, and vegetation growth; blue arrows, research concept: (1) Can the CRSbAI replace CdSWC? (2) Is the CRSbAI related to vegetation growth?

*2.3. Effective Cumulative Reciprocal SbAI*

ECT is calculated as follows:

$$ECT \ = \ \sum (T_d - T_b), \tag{4}$$

where $T_d$ is the daily averaged temperature (°C) and $T_b$ is threshold temperature (°C) determined by the plant species.

Numerous studies report the relationships among rainfall, SWC, and vegetation growth (Figure 2b, red arrows) in natural grasslands of north-east Asia [9,19,43,47,48]. Starting with the ECT method, we replaced temperature with the effective cumulative reciprocal of SbAI (CRSbAI) obtained from satellite data (Figure 2b) to monitor the SWC consumption and vegetation growth as follows:

$$CRSbAI \ = \ \sum_{k=i}^{k=j} \left( \left[ SbAI^{-1} \right]_k - 1/0.03 \right), \tag{5}$$

where $SbAI^{-1}$ is the reciprocal of SbAI, because SbAI decreases with increasing thermal inertia; $i$ is the initial month of growth (March in Shenmu, April in Bayan-Unjuul); $j$ is the month when growth reaches the maximum (August at both sites); $k$ is month; and 0.03 is the threshold value of the SbAI in extreme dry conditions in which wind erosion is likely [6,38].

As the SbAI cannot always be derived owing to the effect of cloud cover, we used a 16-day moving average to obtain the continuous, daily SbAI.

This study asked the following two questions (Figure 2b, blue arrows):

1.    Can the CRSbAI replace the cumulative decreased SWC (CdSWC)?
2.    Is the CRSbAI related to vegetation growth?

We developed these concepts at Shenmu and verified them at Bayan-Unjuul. Indices which represent vegetation biomass should be based on observed above-ground dry matter. However, we used the NDVI, which is strongly correlated with the observed dry matter in grasslands of north-east Asia [48–50].

## 3. Results

### 3.1. Seasonal Change in the NDVI, SWC, and SbAI in Shenmu

In Shenmu, the NDVI increased gradually to a maximum of 0.35 to 0.4 in August (Figure 3). The total rainfall from March to August was 215 mm in 2005, 309 mm in 2006, and 303 mm in 2007. Although the total rainfall was higher in 2007 than in 2005, the NDVI was lowest in August 2007. This was caused by rainfall intensity and thus a higher SWC in June to July (DOY 152 to 212) (Figure 4). In June to July, the SWC approached field capacity ($0.19 \ m^3 \ m^{-3}$) and stayed high in 2005 and 2006, but was lower ($<0.15 \ m^3 \ m^{-3}$) in 2007. Concentrated rainfall in June to July strongly increased subsequent vegetation biomass in the grasslands of Mongolia and China [19,47].

Although the seasonal change in the daily SbAI varied, the trend of the 16-day moving-average SbAI mirrored that of the SWC (one was high when the other was low; Figure 4). The SbAI had a significant correlation with SWC (r = 0.405, $p < 0.05$). Here, r is the correlation coefficient, and $p$ is the $p$-value.

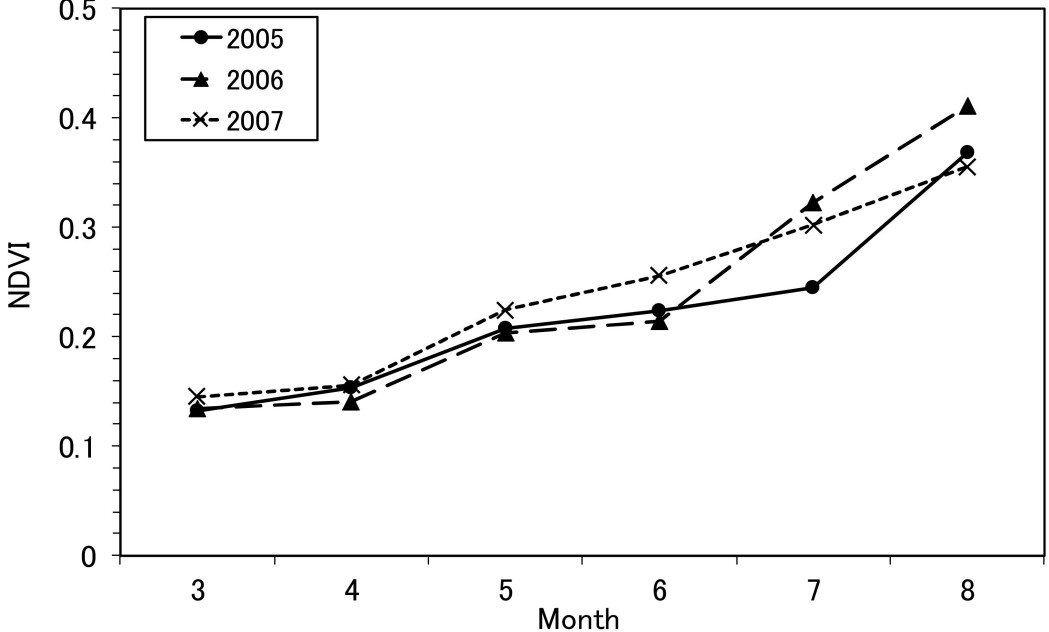

**Figure 3.** Monthly NDVI in Shenmu in 2005 to 2007.

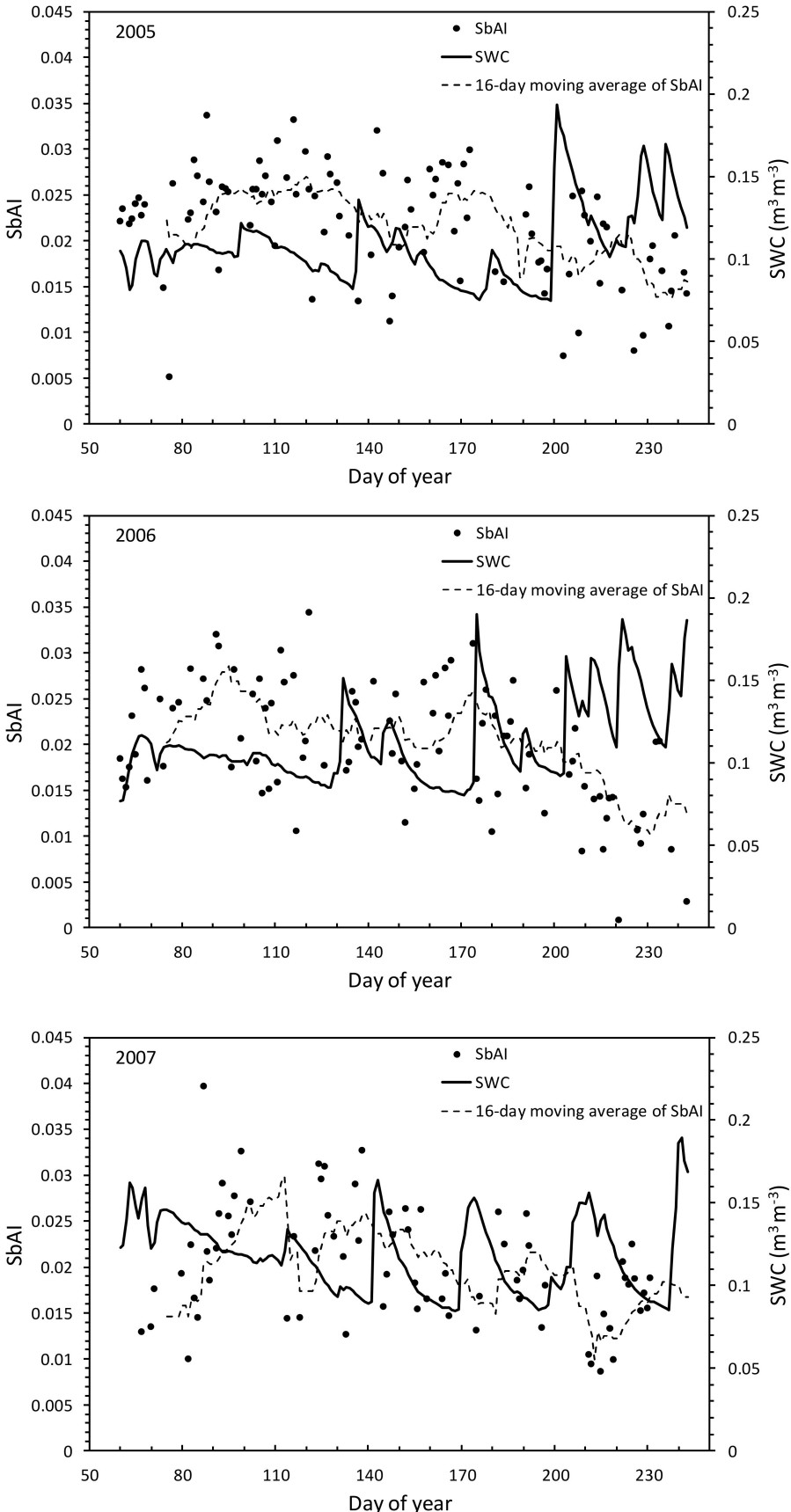

**Figure 4.** Seasonal change in the SbAI and SWC averaged within the root zone in Shenmu in 2005 to 2007.

### 3.2. Relationships among CRSbAI, CdSWC, and $\Delta_{NDVI}$ in Shenmu

The relationships of CRSbAI (daily) with CdSWC within the root zone and $\Delta_{NDVI}$ were examined in Shenmu from 2005 to 2007 (Figure 5). Here, "CRSbAI (daily)" means the CRSbAI from March to March, to April, to May, to June, to July, and to August, calculated from the daily SbAI with the 16-day moving average. The CdSWC covers March to March, to April, to May, to June, to July, and to August. $\Delta_{NDVI}$ is the NDVI in March, April, May, June, July, and August minus the NDVI in February, the month with no grass growth.

The CRSbAI was significantly correlated with the CdSWC (r = 0.984, RMSE = 8.7 mm, $p < 0.001$) and $\Delta_{NDVI}$ (r = 0.966, RMSE = 0.022, $p < 0.001$). Here, RMSE is the root-mean-squared error. Although the rainfall amount and intensity differed among the years, CRSbAI integrated them into a single linear relationship. An index of thermal inertia based on LST was showed to be most highly related to SWC within the root zone during grass growth in China [51]; they concluded that wet soil may cause both high transpiration and evaporation at the ground surface. These findings and our results in Figure 5 may validate the concept shown in Figure 2.

The final CdSWC values were 147 mm in 2005, 169 mm in 2006, and 142 mm in 2007 (Figure 5, blue arrows). The CdSWC is the evapotranspiration (ET), and its ratio to the total rainfall during study period was 68% in 2005, 55% in 2006, and 47% in 2007. These ratios are close to the results obtained in past studies in grasslands of California (54–59%; [52]), South Africa (40–76%; [53]), and Mongolia (66%; [54]).

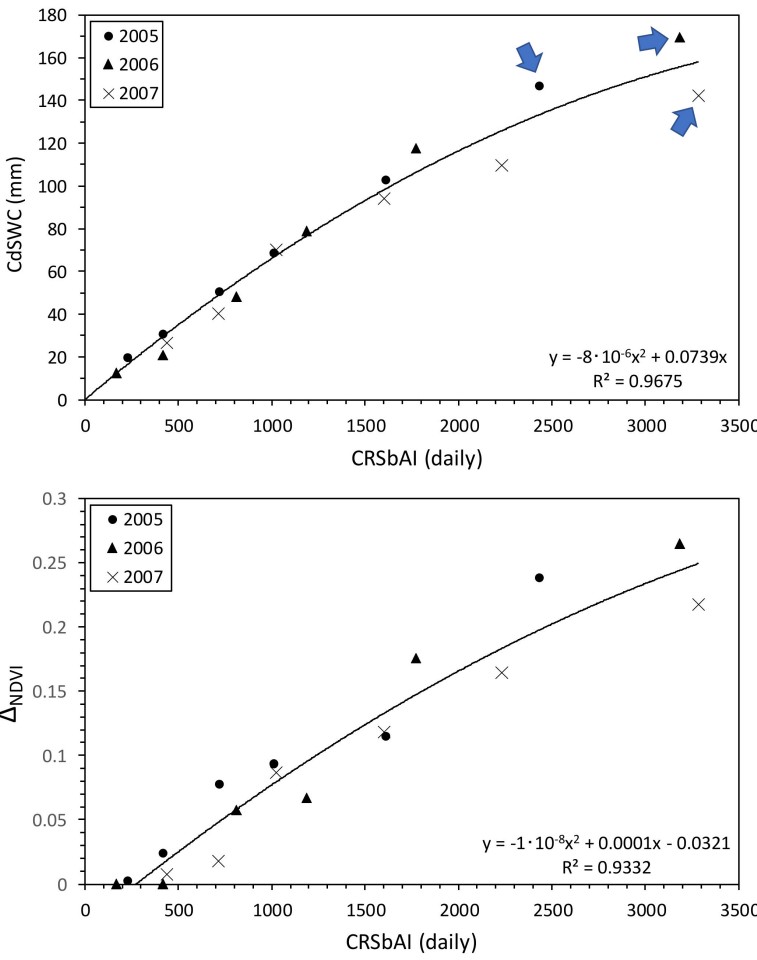

**Figure 5.** Relationships of CRSbAI (daily) with CdSWC and $\Delta_{NDVI}$ in Shenmu from 2005 to 2007. Blue arrows show the final CdSWC values.

The CRSbAI (monthly) was also significantly and highly correlated with CdSWC (r = 0.981, RMSE = 9.4 mm, $p < 0.001$) and $\Delta_{NDVI}$ (r = 0.966, RMSE = 0.022, $p < 0.001$) (Figure 6). Here, the CRSbAI (monthly) was calculated from the monthly average observed SbAI. Therefore, the CRSbAI (monthly) can be calculated easily from the observed data without statistical processing (such as a 16-day moving average).

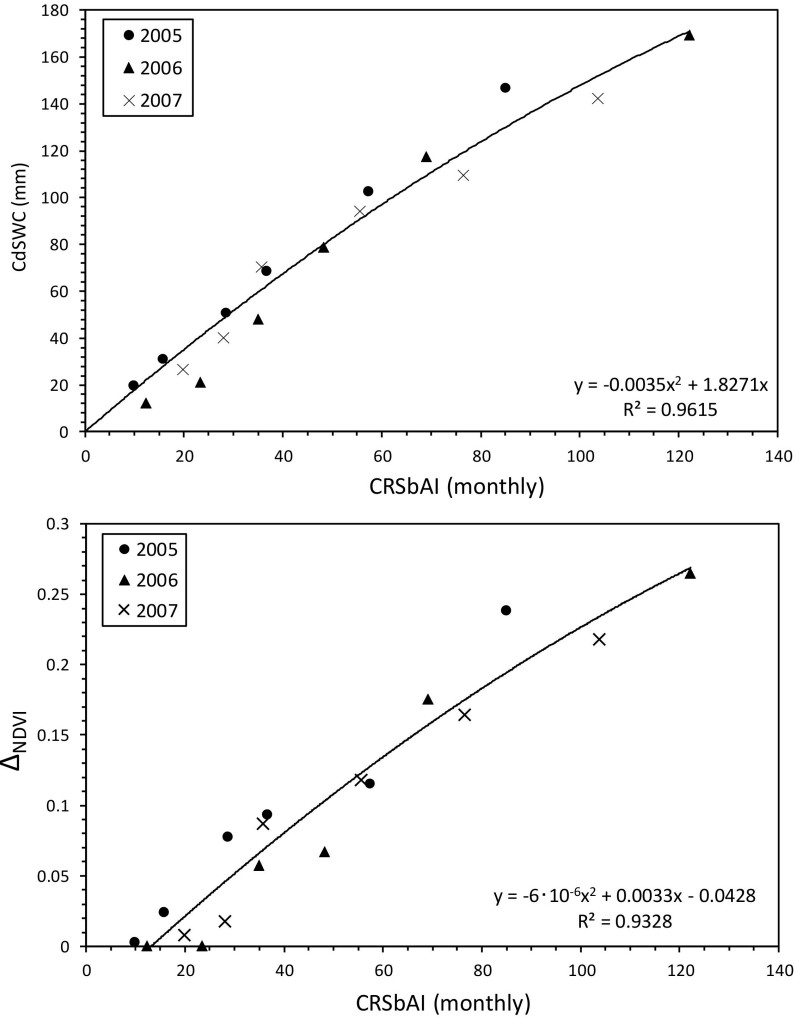

**Figure 6.** Relationships of CRSbAI (monthly) with CdSWC and $\Delta_{NDVI}$ in Shenmu from 2005 to 2007.

### 3.3. Validation of the CRSbAI Method in Bayan-Unjuul

In Bayan-Unjuul, the seasonal NDVI was steady from March to June and then increased rapidly to August (Figure 7). This is a general trend in grass growth in Mongolia, caused by the rainfall concentrated in June and July [19,47]. In Shenmu, however, the NDVI increased slowly from March to August (Figure 3) because of frequent rainfalls. This is a crucial difference in grass growth between the sites. The severe drought in 2017 [44] explains the lower NDVI in July and August 2017 when compared to 2018 and 2019.

Both the CRSbAI (daily) and CRSbAI (monthly) agreed well with $\Delta_{NDVI}$ (daily: r = 0.982, RMSE = 0.024, $p < 0.001$) (monthly: r = 0.960, RMSE = 0.030, $p < 0.001$) (Figure 8), as at Shenmu (Figures 5 and 6). The calculation began in April (March in Shenmu) because the soil was still frozen in March. From April to June, when the rainfall was low, the small CRSbAI corresponded to the small $\Delta_{NDVI}$. From July, however, during the rainy season, as the CRSbAI increased rapidly, so too did the $\Delta_{NDVI}$. This validation result in Bayan-Unjuul indicates that the CRSbAI responds well to the water condition of the land surface assumed in Figure 2a.

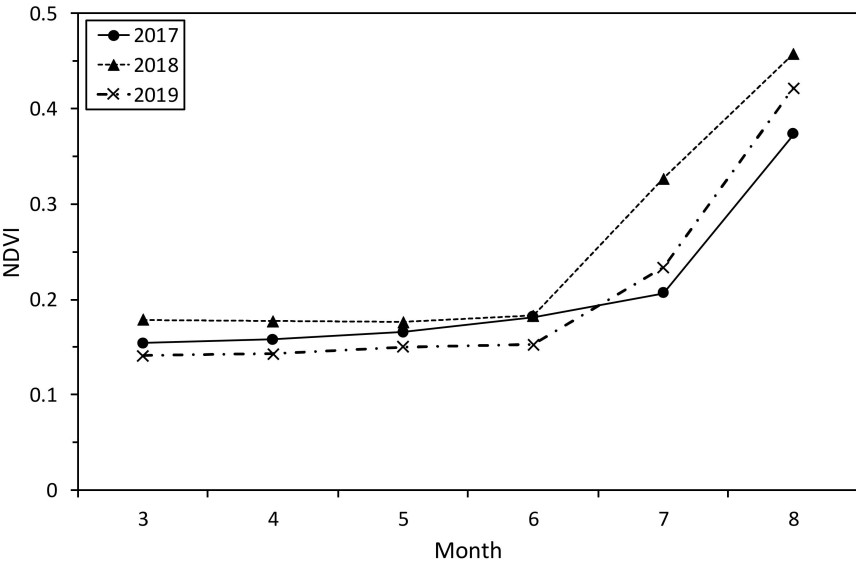

**Figure 7.** Monthly NDVI in Bayan-Unjuul from 2017 to 2019.

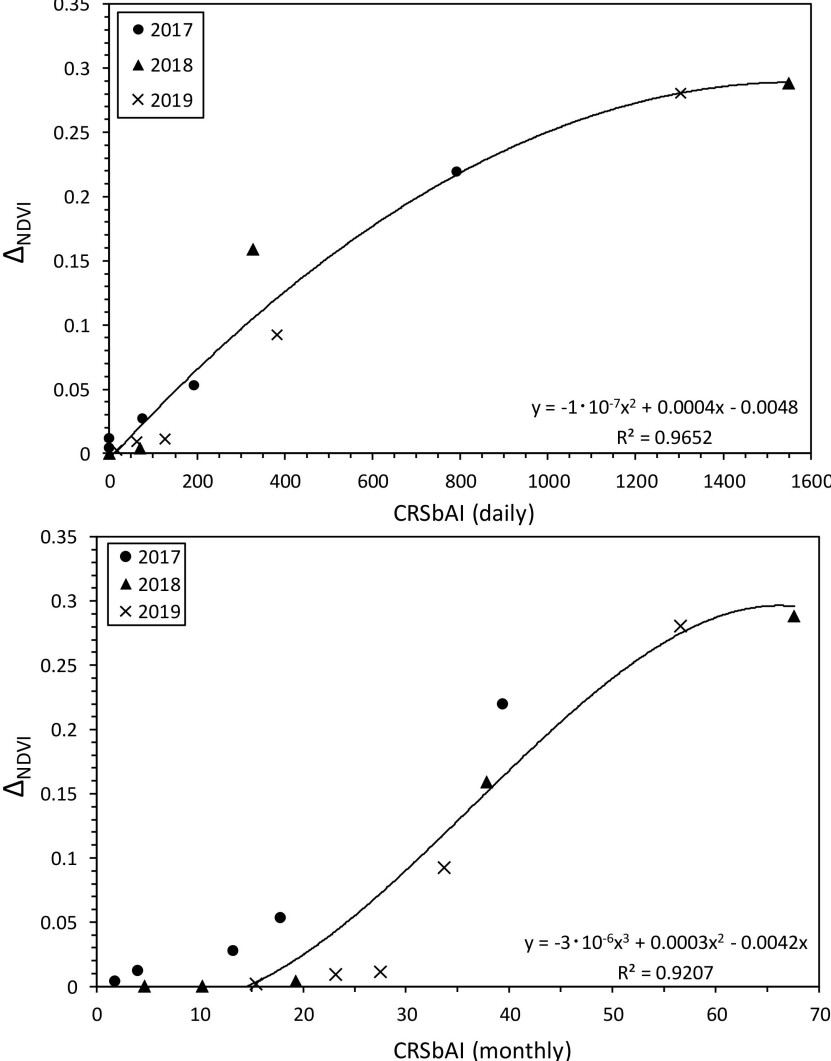

**Figure 8.** Relationships of CRSbAI (daily) and CRSbAI (monthly) with $\Delta_{\text{NDVI}}$ in Bayan-Unjuul from 2017 to 2019.

$\Delta_{NDVI}$ peaked at around 0.3 in Bayan-Unjuul (Figure 8); the corresponding CRSbAI values were 1300 to 1600 daily and 50 to 70 monthly. Although the $\Delta_{NDVI}$ peaked at around the same value in Shenmu (Figures 5 and 6), the corresponding CRSbAI values were 3000 to 3500 daily and 100 to 130 monthly, double those in Bayan-Unjuul. This means that the grass in Bayan-Unjuul has a high water-use efficiency by fitting its physiological activity to the late rainfall.

### 3.4. Simulation of CdSWC in Bayan-Unjuul

As the sandy loam and dominant grass species were comparable between Shenmu and Bayan-Unjuul, we assumed that the only serious difference between the sites was rainfall. The daily and monthly CdSWC values in Shenmu (Table 1) were calculated as follows:

$$CdSWC_{(daily)} = -8 \cdot 10^{-6}CRSbAI^2_{(daily)} + 0.0739 \cdot CRSbAI_{(daily)} \ 0 < CRSbAI_{(daily)} < 3500 \qquad (6)$$

$$CdSWC_{(monthly)} = -0.0035 \cdot CRSbAI^2_{(monthly)} + 1.8271 \cdot CRSbAI_{(monthly)} \ 0 < CRSbAI_{(monthly)} < 140. \quad (7)$$

**Table 1.** The CdSWC to August calculated from the CRSbAI and Equations (6) and (7) with the $\Delta_{NDVI}$ (August minus March) in Bayan-Unjuul from 2017 to 2019.

|  | $\Delta N_{DVI}$ | CRSbAI (Daily) | CRSbAI (Monthly) | CdSWC (mm) (Daily) | CdSWC (mm) (Monthly) |
|---|---|---|---|---|---|
| 2017 | 0.219 | 794 | 39 | 54 | 66 |
| 2018 | 0.279 | 1549 | 68 | 95 | 108 |
| 2019 | 0.281 | 1302 | 57 | 83 | 93 |

The daily and monthly values were close, differing by only 10 mm among years.

The CdSWC was lowest in 2017 (54 to 66 mm), owing to drought, and that in 2018 was highest (95 to 108 mm), owing to heavy rainfall [45]. As the CdSWC (monthly) is the ET, the ET was 66 mm in 2017, 108 mm in 2018, and 93 mm in 2019. The Institute of Meteorology, Hydrology, and Environment records rainfall in Bayan-Unjuul. From April to August it recorded 122 mm in 2017, 154 mm in 2018, and 159 mm in 2019. On the Mongolian dry steppe near Bayan-Unjuul (Kherlenbayan-Ulaan: 47°12.838′ N, 105°44.240′ E, 1235 m above sea level), the ratio of ET to precipitation is 66% [54]. At that site, the soil is a Kastanozem and the predominant vegetation is *S. krylovii*, the same as at Byan-Unjuul. When this ratio was used, the ET was calculated as 81 mm in 2017, 102 mm in 2018, and 105 mm in 2019, close to the ET as CdSWC (root-mean-squared error of ±11 mm)

### 4. Discussion

Here, we will discuss the applicability evaluation of the CRSbAI over north-east Asia from 2017 to 2019, the three most recent years; that is, the spatial distribution of the CRSbAI and CdSWC will be examined using the calculation method obtained in Shenmu and Bayan-Unjuul, and validated using past published data.

Figure 9 maps the distributions of the CRSbAI (monthly) (cumulated from April to August) and NDVI in August from 2017 to 2019. Although the response of $\Delta_{NDVI}$ to the CRSbAI differed among years (Figures 5, 6 and 8), the distribution of the CRSbAI corresponded well with that of the NDVI:

NDVI < 0.2 ≡ CRSbAI < 40;
0.2 < NDVI < 0.4 ≡ 40 < CRSbAI < 80;
0.4 < NDVI < 0.6 ≡ 80 < CRSbAI < 120.

In particular, the Gobi Desert, south central Mongolia, Inner Mongolia, the Loess Plateau of China, and irrigated farmland along the Yellow River were well delineated by the CRSbAI. The area where the CRSbAI < 60 (purple) was large in 2017, when severe drought prevailed, and the area

with a CRSbAI ≈ 80 (blue) increased in 2018 owing to the heavy rainfall. Thus, the CRSbAI reflected differences in rainfall intensity.

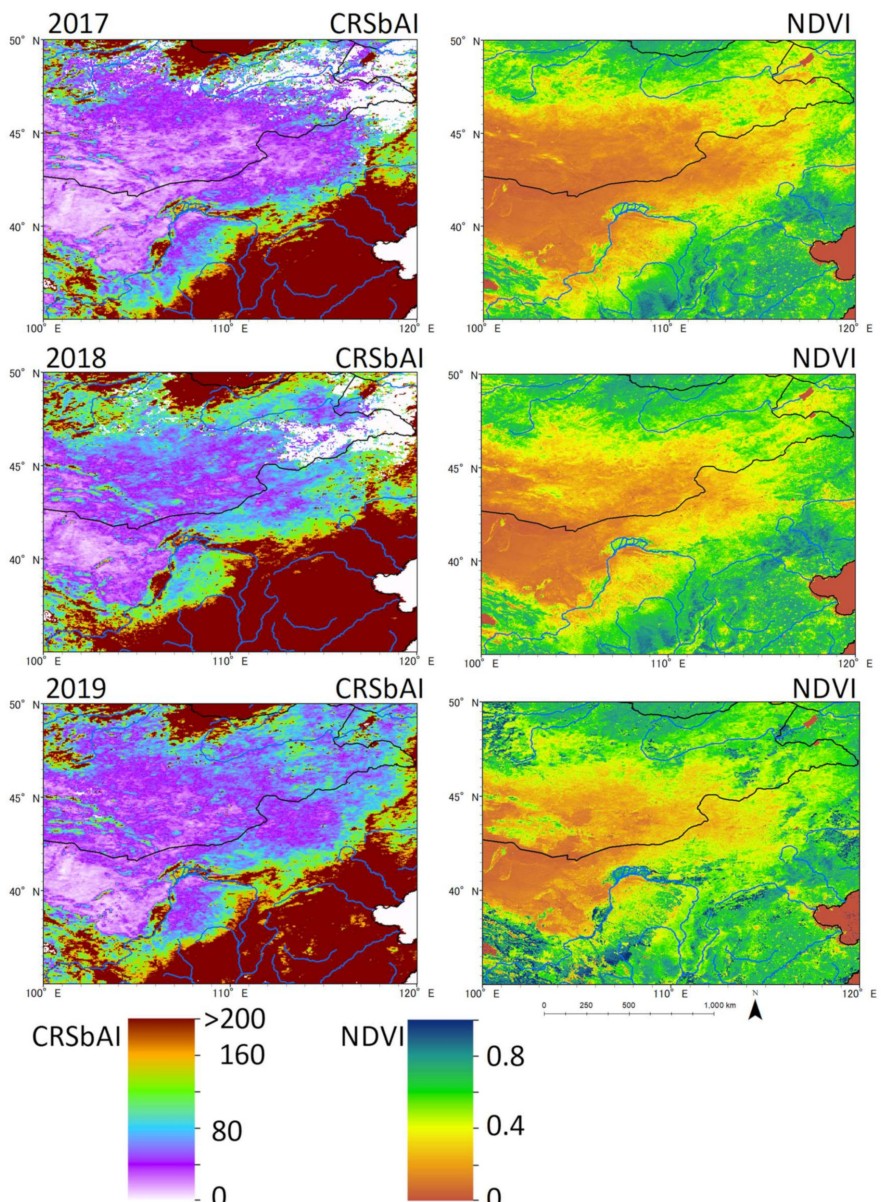

**Figure 9.** Spatial distributions of the CRSbAI (monthly) and NDVI in August in north-east Asia from 2017 to 2019. White areas indicate where cloud cover prevented calculation.

Figure 10 shows the distribution of CdSWC calculated with Equation (7) and the total precipitation (Pr) from April to August 2017 to 2019, using the CPC Global Unified Precipitation data. Since the range of CRSbAI (monthly) was approximately 0 to 150 mm (Figure 6), areas where the CRSbAI were >150 mm were not available. However, areas where the CRSbAI were <150 mm corresponded closely to the Pr distribution:

$0 < \text{CdSWC} < 50 \equiv 0 < \text{Pr} < 80;$
$50 < \text{CdSWC} < 70 \equiv 80 < \text{Pr} < 130;$
$70 < \text{CdSWC} < 110 \equiv 130 < \text{Pr} < 210;$
$110 < \text{CdSWC} \equiv 210 < \text{Pr}.$

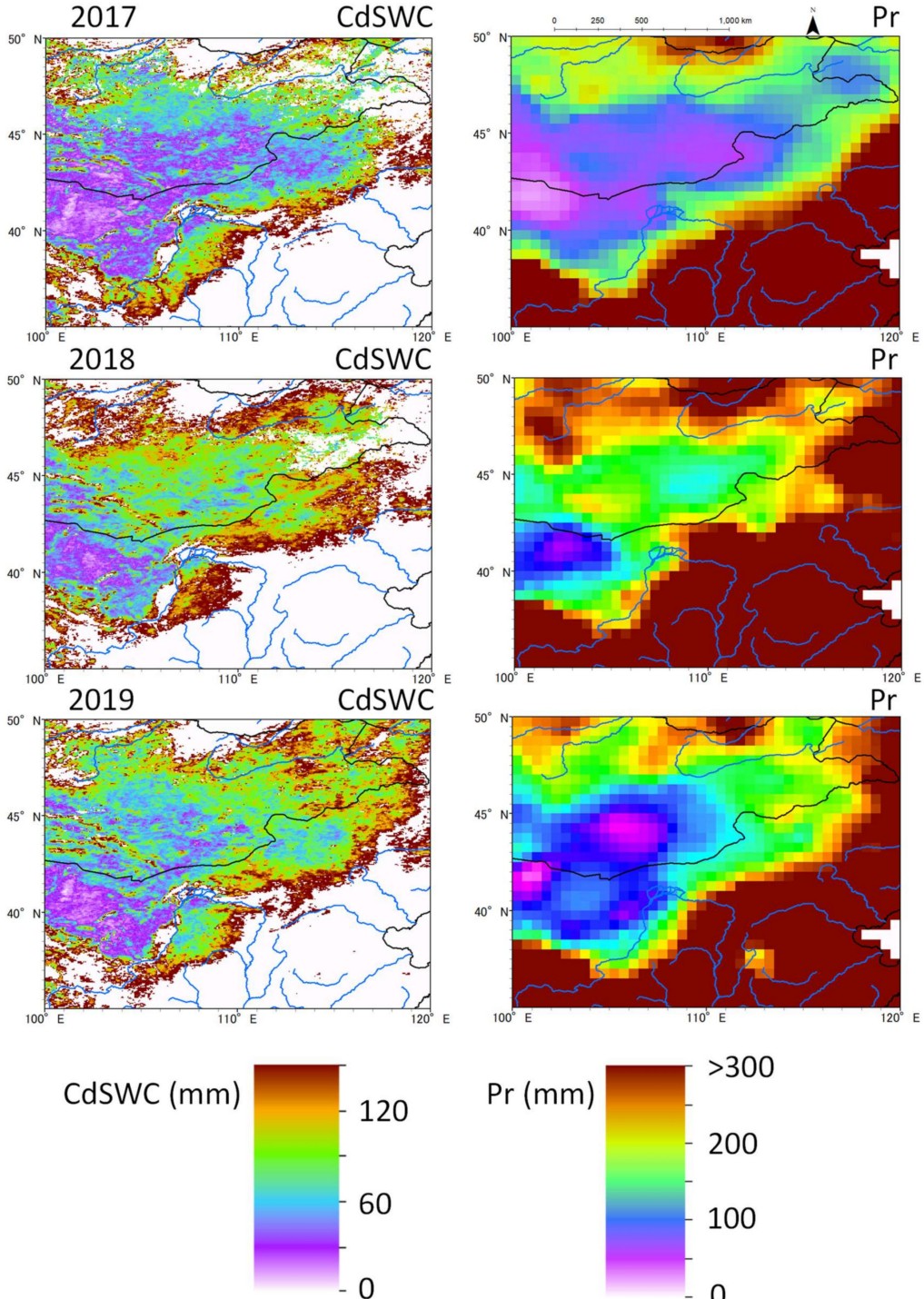

**Figure 10.** Spatial distribution of the CdSWC (mm) calculated from the CRSbAI (monthly) and precipitation (Pr, mm) from April to August 2017 to 2019, using the CPC Global Unified Precipitation data. White areas indicate where the CRSbAI > 150 mm (therefore, data not available).

It is of note that the distribution of the CdSWC corresponded closely to the arid or semi-arid regions [42,55], mostly the Gobi Desert and natural grasslands where dust outbreaks arise [56–58].

As the CdSWC is the ET, the ET in the Gobi Desert was <30 mm during April to August. The ET was sensitive to rainfall in south-central Mongolia, Inner Mongolia (China), and the Loess Plateau of China (Figure 10). Drought in 2017 caused a low ET and NDVI, not only over a wide area in Mongolia,

but also in Inner Mongolia. Thus, we can easily predict that such an effect of drought can trigger outbreaks of dust in these areas.

It is not clear whether Equation (7) can be applied to all areas in Figure 10. However, the ET observed by the eddy covariance method from May to August 2014 in Tsogt-Ovoo (desert steppe; 44°23′04″ N, 105°16′59″ E, 1232 m above sea level) is one reason for the validation [59]. The annual rainfall in 2014 was 72 mm and normal (the average from 1981 to 2010 was 75 mm), and the observed ET was 50 mm. This value is close to the ET values of the cell including this point (23 mm in 2017, 75 mm in 2018, and 38 mm in 2019). Another validation result was reported by [54], who calculated the ET by the eddy covariance method in Kherlenbayan-Ulaan, Mongolia, in 2003; the average annual rainfall from 1993 to 2002 was 181 mm, and the value in 2003 (248 mm) was larger than normal; the observed ET was 120 mm from April to August, only slightly higher than our CdSWC value of 103 mm in 2018, when the rainfall was heavier than normal.

The ratio of the CdSWC to Pr was 52% to 63% across their spatial distribution (Figure 10), close to the values in the dry steppe [54] and other grasslands [52,53].

## 5. Conclusions

We used a satellite-based aridity index (SbAI) calculated only from satellite data, the effective cumulative reciprocal SbAI (CRSbAI), to monitor the SWC and grass growth in north-east Asia, calibrating it in natural grasslands in Shenmu, China, and validating its applicability in Bayan-Unjuul, Mongolia. We asked the following questions:

- Can the CRSbAI replace CdSWC?
- Is the CRSbAI related to grass growth and represented by $\Delta_{NDVI}$?

The CRSbAI was significantly correlated with CdSWC in Shenmu. Although the rainfall differed among years, the CRSbAI integrated the rainfall values in a single linear relationship. As the CdSWC calculated from the CRSbAI is the ET, the ET in Bayan-Unjuul was 66 mm in 2017, 108 mm in 2018, and 93 mm in 2019. As the ET is 66% of the precipitation in the Mongolian steppe, the estimated ET was 81, 102, and 105 mm, respectively. These respective values are close (the root-mean-squared error is within ±11 mm). The ratio of CdSWC to Pr across north-east Asia was 52% to 63%, close to the ratios in other grasslands reported in past studies.

The CRSbAI was strongly related to $\Delta_{NDVI}$ at both Shenmu and Bayan-Unjuul, although grass growth responded differently owing to differences in rainfall intensity; also, the spatial distribution of the CRSbAI corresponded well to that of the NDVI in August. This result suggests that the CRSbAI can be used to predict the distribution of the NDVI qualitatively. To predict the $\Delta_{NDVI}$ quantitatively, their relationship should be examined in the respective target regions.

Verification of our assumptions (above two questions) is not yet complete. However, the CRSbAI could offer an index of water consumption and grass growth, and may be useful for development of early warning and monitoring systems for drought. We hope that the usefulness of our method will be confirmed by other researchers, and that it will serve as the basis for an improved system based on remote sensing techniques that will promote sustainable development in north-east Asia.

**Author Contributions:** Conceptualization, R.K.; methodology, R.K.; software, M.M.; validation, R.K. and M.M.; formal analysis, R.K.; resources, M.M.; data curation, R.K. and M.M.; writing—original draft preparation, R.K.; writing—review and editing, R.K.; visualization, R.K.; supervision, R.K.; project administration, R.K.; funding acquisition, R.K. All authors have read and agreed to the published version of the manuscript.

**Funding:** This research was funded by a Grant-in-Aid for Scientific Research, grant number KAKENHI 19H04239.

**Acknowledgments:** We sincerely thank Amarsaikhan Davaadorj of the Institute of Meteorology, Hydrology, and Environment, Mongolia, and Batjargal Buyantogtoh of Tottori University for preparing the precipitation data from Bayan-Unjuul. We appreciate the invaluable comments from five reviewers and the academic editor of this paper.

**Conflicts of Interest:** The authors declare no conflict of interest. The funders had no role in the design of the study; in the collection, analyses, or interpretation of data; in the writing of the manuscript, or in the decision to publish the results.

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
