# Peer review of "Use of a Satellite-Based Aridity Index to Monitor Decreased Soil Water Content and Grass Growth in Grasslands of North-East Asia"

_remotesensing, doi:10.3390/rs12213556_

Round 1
Reviewer 1 Report
The article points to the possibilities of using remote sensing in the assessment of the climatic characteristics of the territory. remote sensing is a tool for processing large amounts of data and interpreting them directly. The results point to new possibilities of using the aridity index for qualitative NDVI determination, and in Shennu correlates with the SWC. ACSbAI can be useful for early warning and monitoring systems for droughts.
One question: in China, the installation and collection of climate and pedological data from many measuring instruments are described. Were the same devices used in Mongolia? If not, why?
Author Response
Dear Reviewer 1,
I do appreciate for your review and taking your precious time. Please see the attachment to your comments.
sincerely,
Reiji Kimura

Reviewer 2 Report
The authors used MODIS data to develop the aridity index (SbAI) and effective cumulative reciprocal SbAI (ECSbAI) and applied the reciprocal index as a way to measure water consumption and growth of grasslands in China and Mongolia. The data presented showed a strong and clear relationship between ECSbAI and soil water content and NDVI, thereby forming an effective calibration. The regional and continental-scale (spatial) numerical simulations are sufficiently detailed so as to identify arid or semi-arid regions, and their soil moisture and potential for grass growth. This approach will be very valuable in grassland management and also understanding the large-scale patterns of soil moisture limitation with increasing climate change.
Overall, I very much enjoyed this manuscript. The idea of the index is expressed extremely well, there are correlated measures and robust ground truthing, and we get a solid understanding of the system moisture dynamics with fewer data points. If there is one thing that I can comment about it is the way the authors use unnecessarily complicated acronyms. The structure of the acronyms in some instances look like chemical formulae. They don’t seem to reflect the underlying properties they describe as well as they could. Good acronyms provide hints to the underlying properties. However, I found it necessary to back trace throughout the manuscript to reference the actual names of the properties more frequently than usual, interrupting the flow of the otherwise excellent paper, thus making it harder to understand.
Author Response
Dear Reviewer 2,
I do appreciate for your review and taking your precious time. Please see the attachment to your comments.
sincerely,
Reiji Kimura

Reviewer 3 Report
Dear authors,
The paper is very well written and organized. It presents new data and a very relevant topic which is the problem to measure water consumption and the growth of grasslands in China and Mongolia.
I have minor editing comments:
All indications of the Figures in the text are presented in red and they should be changed to black.
Author Response
Dear Reviewer 3,
I do appreciate for your review and taking your precious time. Please see the attachment to your comments.
sincerely,
Reiji Kimura

Reviewer 4 Report
Dear Authors,
I have reviewed the paper "Use of a satellite-based aridity index to monitor decreased soil water content and grass growth in grasslands of north-east Asia". The aims of the paper are germane with Remote sensing applications topic, in this form of article fits with the international scientific standards. The paper is written with a moderate English level. The contribution of this paper to the scientific knowledge is good. Although, in my opinion, there some important flaws and I suggest the corrections in the comments in the attached file.

Author Response
Dear Reviewer 4,
I do appreciate for your review and taking your precious time. Please see the attachment to your comments.
sincerely,
Reiji Kimura

Reviewer 5 Report
See attached comments

Author Response
Dear Reviewer 5,
I do appreciate for your review and taking your precious time. Please see the attachment to your comments.
sincerely,
Reiji Kimura

Round 2
Reviewer 4 Report
Good job...
Author Response
Dear Reviewer 4,
I do appreciate for your warm comments.
Sincerely yours,
Reiji Kimura
Associate Professor
Arid Land Research Center, Tottori University
Reviewer 5 Report
I've gone through the revised paper version, and I appreciate the efforts of the authors to improve their work. I think the paper is now ready for publication.
Author Response
Dear Reviewer 5,
I do appreciate for your warm comments.
Sincerely yours,
Reiji Kimura
Associate Professor
Arid Land Research Center, Tottori University